# cfDNA Chimerism and Somatic Mutation Testing in Early Prediction of Relapse After Allogeneic Stem Cell Transplantation for Myeloid Malignancies

**DOI:** 10.3390/cancers17040625

**Published:** 2025-02-13

**Authors:** Scott D. Rowley, Maher Albitar, Melissa F. Baker, Alaa Ali, Sukhdeep Kaur, Hyung C. Suh, Andre Goy, Michele L. Donato

**Affiliations:** 1John Theurer Cancer Center, 92 Second St., Hackensack, NJ 07601, USA; melissa.baker@hmhn.org (M.F.B.); sukhdeep.kaur@hmhn.org (S.K.); hyung.suh@hmhn.org (H.C.S.); goy.andre@hmhn.org (A.G.); michele.donato@hmhn.org (M.L.D.); 2Lombardi Comprehensive Cancer Center, Georgetown University School of Medicine, 3800 Reservoir Rd. NW, Washington, DC 20057, USA; alaa.ali@gunet.georgetown.edu; 3Genomic Testing Cooperative, 25371 Commercentre Dr., Lake Forest, CA 92630, USA; malbitar@genomictestingcooperative.com

**Keywords:** cell-free DNA, next-generation sequencing, allogeneic hematopoietic cell transplantation, post-transplant relapse

## Abstract

Relapse of disease is a primary cause of treatment failure after allogeneic bone marrow transplantation. Patients may benefit from post-transplant consolidation therapy, but such therapy poses serious risks requiring careful balancing with the potential benefits. We performed a pilot study detecting disease-associated somatic mutations by measuring cell-free DNA (cfDNA) in the plasma after transplantation. We found that clearance or persistence of adverse-risk defining mutations tested as early as 28 days after transplantation identified patients at low or high risk of relapse. These data support the hypothesis that testing cfDNA early after transplantation will identify individual patients who may benefit from early modification of their treatment plans to reduce the risk of relapse. Testing for cfDNA may be also useful in the design and interpretation of clinical protocols testing various conditioning and GvHD regimens and post-transplant consolidation techniques in which disease relapse is a primary or secondary endpoint.

## 1. Introduction

Disease relapse is the primary cause of treatment failure after allogeneic hematopoietic stem cell transplantation (HSCT) in the treatment of malignancy. The risk of relapse is associated with various patient, disease, and treatment factors, including the presence of adverse-risk somatic mutations detected at the time of transplantation [1,2,3,4]. Consolidation therapy early after transplantation may reduce the risk of relapse, but most diseases lack a treatment target and require the use of non-specific approaches such as chemotherapy, early withdrawal of immunosuppression, or prophylactic or preemptive infusion of donor lymphocytes (DLI) [5,6,7]. Unfortunately, each of these approaches increases the risk of treatment-related toxicities and may not be suitable if the patient is experiencing a post-transplant course complicated by poor graft function, opportunistic infections, or graft-versus-host disease (GvHD).

The ability to detect and monitor residual disease (MRD) early after transplantation would be advantageous if such testing could accurately identify those patients who would benefit from modifications to each individual patient’s treatment plan, effectively reducing the risk of relapse and treatment failure. Recently, techniques for detecting tumor-derived circulating cell-free DNA (cfDNA) have been developed. The advantages of a “liquid biopsy” include the rapid clearance of plasma cfDNA within hours; allowing sampling early after treatment; possibly more accurate identification of residual disease burden compared with otherwise patchy bone marrow involvement; the ability to study the clonal evolution of disease; and avoidance of an invasive sampling technique, thereby facilitating serial measurements [8,9,10,11,12]. Furthermore, measuring plasma cfDNA allows us to evaluate various single-nucleotide polymorphisms (SNPs) to determine the donor chimerism level of hematopoietic cells [13].

We hypothesized that detection of plasma cfDNA for identifying MRD and its associated measurement of donor chimerism during the first 12 weeks after allogeneic HSC transplantation may be clinically useful in identifying patients likely to relapse, allowing intervention at an earlier stage of the patient’s transplant course when such treatment may be more effective. We performed a prospective pilot study to determine the ability to detect cfDNA, which may reflect sustained MRD early after transplantation in a cohort of subjects with known pre-transplant somatic mutations undergoing allogeneic transplantation.

## 2. Methods

This is a single-center, non-randomized, prospective pilot study of the feasibility and potential clinical utility of detecting leukemia-associated somatic mutations through NGS analysis of cfDNA early after transplantation. The study’s eligibility criteria included a diagnosis of a myeloid malignancy for which allogeneic HSCT was clinically indicated and with pre-transplant documentation of tumor-associated genetic mutation(s). Subjects were required to meet the transplant eligibility criteria set by the transplant program. The choice of conditioning and GvHD prophylaxis regimens; source of HSC; and recipient or donor age or sex, ABO match, HLA match, and CMV status were not defined by protocol and were per the discretion of the treating physician. Transplant conditioning and GvHD prophylaxis regimens and post-transplant care are described in the Appendix A. Details regarding the enrolled subjects are described in Table 1 and Table 2. Per protocol, the subjects were followed for study purposes for 12 months after transplantation.

The institutional review board approved the study protocol, and all subjects provided informed consent for enrollment into the study. Twenty-one subjects were enrolled in the study. One subject (014) expired of disease-related complications before transplantation and was excluded from analysis.

### 2.1. Bone Marrow and Peripheral Blood Analysis

The day the HSC infusion was completed is defined as day 0 for this study’s purposes. Bone marrow (BM) samples for disease evaluation, including cytogenetic and NGS mutation analyses, were planned per transplant program standard practice within 28 days before the initiation of transplant conditioning and again on or after day 84 after transplantation (Appendix A). Plasma samples for cfDNA analyses were obtained before the initiation of transplant conditioning (baseline) and again at days 28, 56, and 84 after transplantation. Study chimerism analyses using cfDNA analysis were also performed on the three samples obtained after transplantation. Testing was performed by Genomic Testing Cooperative (Lake Forest, CA, USA) using commercially available panels (Liquid Trace^®^ (cfDNA) and Hematology Profile Plus (cell samples)) of 302 genes (Appendix A).

CD3^+^ (lymphoid) and CD15^+^ (myeloid) donor cell chimerism through PCR analyses of the short tandem repeat (STR) markers in peripheral blood (PB) samples enriched for CD3^+^ or CD15^+^ cells were performed per transplant program standard practice on PB samples obtained at 4 week intervals beginning at day 28 and up to day 84. Routine post-transplant BM samples scheduled at day 84 were evaluated for evidence of relapse using standard techniques, including NGS analysis of the cell sample, and for donor chimerism using CD34^+^ cell-enriched samples. The timing of the routine PB cellular chimerism analysis and collection of BM samples could be modified by the physician caring for the patient.

### 2.2. Post-Transplant Consolidation Therapy

Consolidation therapy (Table 2) after transplantation was not defined under the study protocol but was allowed per the decision of the treating physician. Several subjects were candidates for interventional studies of post-transplant consolidation, which required BM samples be obtained earlier than the standard day 84 time point.

### 2.3. Diagnosis of Relapse

Relapse was defined as the persistence or recurrence of disease meeting standard definitions of disease relapse, as documented by BM biopsy or a peripheral blood (PB) sample, and requiring reinitiation of therapy or infusion of donor lymphocytes (DLI). Persistence or recurrence of minimal residual disease (MRD) based on NGS testing of PB or BM samples was not defined as a relapse. The date of relapse was the date the BM biopsy or diagnostic PB sample confirming relapse was obtained.

### 2.4. DNA and RNA Extraction and Sequencing

The details regarding sample collection and analysis are described in the Appendix A.

cfDNA Analysis: Mutation identification and variant allele frequency (VAF) values were based on plasma cfDNA data. (Mutations identified by cfRNA analysis also performed on these samples (Appendix A) were not analyzed or otherwise reported in this manuscript.) Multiple mutations in a single gene were identified for several subjects (Appendix A). Each of these mutations was included in calculations of the geometric mean (GM) for VAFs found at each time point of sampling. We excluded from this analysis presumed germline mutations of a host origin and any attributed to donor transmission.

DNA Chimerism Analysis: Study samples were tested for residual host cfDNA and used for the calculation of donor chimerism (Appendix A). Sixteen different SNPs were selected from the genomic DNA panel. The median difference in the VAF using the informative SNPs for each patient was used to determine the level of chimerism.

### 2.5. Statistical Analysis

Patients’ characteristics were summarized (Table 1 and Table 2) using standard statistical analysis. Continuous variables were summarized with medians and ranges, and categorical variables were summarized with counts and percentages. The Pearson correlation coefficient was used to determine the linear relationship between the chimerism methods. A Kruskal–Wallis test was used to compare the numbers of mutations detected in BM and plasma analyses. A Wilcoxon matched-pair test was used to compare the allele frequency of each mutation detected between the BM and plasma, assigning a value of 0 for mutations which were detected in one specimen but not the other.

Overall survival was defined as the time from day 0 to the day of death. Relapse-free survival was defined as the time from day 0 to the day of either relapse or death.

## 3. Results

Twenty subjects underwent transplantation, with 19 subjects followed through the planned day 84 cfDNA sampling (Appendix A). One subject expired without evidence of relapse of transplant-related complications at day 59 after transplantation, and two others succumbed to infections at days 268 and 357. Three subjects expired due to a relapse of disease at 125, 250, and 285 days after transplantation. The median survival time for all subjects was >365 days at the time of data analysis (study follow-up was 12 months).

Consolidation Therapy: Nine subjects received post-transplant consolidation therapy commencing at a median of 77 days after transplantation (range: 35–121; Table 2) using either targeted (n = 4) or non-targeted (n = 5) techniques. Only one subject (009) was treated in a research study exploring modified DLI.

Relapse of Disease: Six subjects relapsed at a median of 153 days (range: 52–170; Table 2). Three of these subjects expired, but three subjects remained alive at >365 days at the time of data analysis.

Chimerism: Routine day 28 donor CD3^+^ cell chimerism analyses were obtained for all subjects, as well as at day 56 for 18 subjects and day 84 for 15 subjects (Appendix A). The median day 28, 56, and 84 donor CD3^+^ cell chimerisms were 86% (range: 15–100), 82% (range: 18–100), and 88% (range: 15–100), respectively. Post-transplant BM samples were obtained at a median of 79 days (range: 51–110), with samples not obtained for the one subject who expired at day 59. The median donor CD34^+^ cell chimerism was 95% (range: 3–100; Appendix A).

The median donor chimerisms for the cfDNA analysis were 95.5% (range: 60–99), 90% (range: 40–99), and 84% (range: 12–97) for testing at days 28, 56, and 84, respectively (Appendix A). We found no correlation between the donor chimerism determined by cfDNA analysis and the CD3^+^ cell (or CD15^+^ cell) chimerism determined by STR analysis at each time point nor for day 84 NGS and BM donor CD34^+^ cell chimerism in the day 84 post-transplant sample (Appendix A).

However, the pattern of cfDNA chimerism appeared to more accurately reflect relapses than the donor CD3^+^ cell chimerism. Two subjects (002 and 010, Appendix A) showed a drop of >50% in donor DNA detected in the NGS analysis, which was consistent with relapses. All relapsed patients showed a constant downward trend in cfDNA chimerism (Figure 1B,D). In contrast, CD3^+^ cell chimerism did not show the same trend (Figure 1A,C).

### 3.1. Concordance of cfDNA and Bone Marrow Samples

Table 3 shows mutations detected in BM samples obtained pre-transplant and post-transplant compared with mutations detected in plasma cfDNA at the same time points. (Complete lists of all mutations detected (including germline mutations) are shown on Appendix A.) Significantly more mutations (Table 3, *p* < 0.0001, Kruskal–Wallis test) were detected in the cfDNA than in the bone marrow cells in pre-transplant testing (92 versus 61 mutations), most likely reflecting sampling variation when bone marrow was used. Subjects with MDS or MPN were more likely to have greater numbers of detectable mutations in cfDNA testing. The median VAF of mutations in the pre-transplant bone marrow samples was 7.53% versus 3.02% in cfDNA (*p* = 0.02) (Wilcoxon matched-pair test). After transplantation, both fewer mutations and lower VAF values were detected, consistent with the clinical response to the initial transplant procedures (Table 2). Overall, 36 mutations were detected in the BM samples, and 36 mutations were detected in the cfDNA analysis. The median VAF was 0.70% in the post-transplant BM cells versus 0.39% in the cfDNA (*p* = 0.13). We note that BM testing was not consistently performed per physician discretion on day 84 when plasma cfDNA testing was performed, which may have degraded the correlation between these two samples.

Table 2 shows the VAF geometric means of mutations obtained in BM or plasma samples at days 28, 56, and 84. All subjects showed a drop in the number of mutations and VAF values compared with the pre-transplant cfDNA samples. However, most subjects with mutations at day 28 showed persistent mutations, with sustained and stable VAF values in the subsequent testing at days 56 and 84. Our data appear to show mutation stability, although the limited number of samples and time course of the study limit this interpretation [14].

### 3.2. Correlation of cfDNA Detection with Relapse

All subjects came to transplantation with somatic mutations identified either at the time of diagnosis or in subsequent pre-transplant NGS testing, per the study’s eligibility criteria (Appendix A). Adverse risk-defining somatic mutations (ASXL1, BCOR, EZH2, FLT3-ITD, RUNX1, SETBP1,SF3B1, SRSF2, STAG2, TP53, U2AF1, ZRSR2, and WT1), as currently defined by the European Leukemia Network (ELN), Pethema Registry, and National Cancer Research Institution (NCRI), were detected before transplantation in all but one (004) of these subjects [14,15,16]. Pre-transplant staging studies for two subjects (008 and 015) were without mutations (MRD-negative) detected in either the pre-transplant BM or plasma samples, and one subject (003) had mutations detected only in the pre-transplant BM sample (Table 2 and Table 3). A fourth subject (021) cleared adverse-risk mutations but was still MRD+ for non-adverse-risk mutations in plasma at day 0.

The testing of cfDNA on days 28, 56, and 84 was without mutations in 6/20, 4/20, and 6/19 evaluable subjects, respectively. Only 1 subject (008) remained without any mutations in all three post-transplant plasma samples tested, while 19 subjects had mutations detected in the plasma at any of the three study time points. The lack of testing at later time points precludes analysis of the durability of clearance of the mutations.

Two subjects (001 and 016) died in remission of regimen-related complications before completing the 12-month follow-up (Table 2). Nine of the 19 subjects with disease characterized by adverse risk-defining mutations at diagnosis or before transplantation had a similar mutation profile detected in plasma samples tested after transplantation, and six of these nine (excluding from this analysis the subject who expired at day 59) experienced early relapses (Table 2 and Table 3, Appendix A, and Figure 2). Four of the subjects (including the two who were MRD-negative in plasma testing) cleared adverse-risk mutations before transplantation, and 5 of the 19 cleared the adverse risk-defining mutations after transplantation. None of these nine subjects suffered relapses.

Several subjects cleared somatic mutations in longer-term, off-study follow-up cfDNA testing (Appendix A), including resolution of TET2 (subject 006), SF3B1 (016), CALR and U2AF1 (019), DMNT3A (020), and CBL (021) mutations. Two of these subjects (016 and 019) cleared adverse-risk mutations and remained in remission. These subjects remained in remission through the end of the study period. The intermittent non-study testing obtained from this limited number of subjects did not allow comprehensive analysis or interpretation of these follow-up data, however.

Both subjects who relapsed before day 100 (subjects 007 and 010) showed a dramatic rise in the VAF GM or number of mutations detected prior to the date of relapse, in association with a fall in donor chimerism (Table 2 and Figure 1D).

## 4. Discussion

Serial measurement of plasma cfDNA via NGS analysis of leukemia-associated somatic mutations during the first 84 days after transplantation demonstrated intermittent or sustained disease markers for most (19 out of 20) subjects undergoing allogeneic PBSC transplantation. These low levels of measurable markers indicative of persistent MRD do not necessarily predict a relapse of disease, although a lack of clearance of mutations of high-risk driver genes appears to be associated with a much higher probability of relapse early after transplantation, possibly justifying modifications in the post-transplant treatment regimen for such patients (Figure 2). In contrast, mutations in epigenetic modifiers and other low-risk mutations did not appear to be as clearly associated with relapse. Our data confirmed the finding of others that failure of clearance of MRD after transplantation is associated with disease relapse (Figure 2) [17,18]. Our data are also consistent with those of others who reported that the genetic profile of the disease at diagnosis is more important in predicting the control of disease than mere MRD positivity at pre-transplant staging (Figure 2) [19,20], possibly explained by distinct genomic aberrations influencing the immunogenicity of malignant cells [21,22,23,24,25].

The detection of adverse-risk somatic mutations in plasma samples obtained after transplantation identified a specific population with a high risk of relapse. If the two subjects (Appendix A) who cleared adverse-risk mutations after the day 84 sampling were included in this analysis, then relapses occurred in six out of seven subjects with sustained adverse-risk mutations. This observation agrees with the NCRI results, demonstrating that classification of patients by adverse-risk mutations in pre-transplant sampling identifies patients predicted to be at a higher risk of relapse after transplantation [16]. The difference in relapses observed in our study, based on the presence or absence of adverse-risk somatic mutations characterizing post-transplant MRD, may be a reflection of particular disease subclones that either persist or are eliminated after transplantation. It is also possible that for the patients with somatic mutations demonstrated by cfDNA analysis but without persisting adverse-risk mutations, we were actually detecting clonal hematopoiesis of indeterminant prognosis (CHIP) present in the individual patients before the development of a myeloid malignancy and not MRD. CHIP may not be associated with the relapse of a myeloid malignancy after transplantation (and conceivably may be eliminated over time through a GvH effect on host hematopoiesis). Larger studies with longer-term testing are required to determine the clinical significance and natural history of persistent detection of non-adverse-risk somatic mutations after transplantation, which would also include the significance of donor-transmitted somatic mutations [22,26,27,28].

The results of this pilot study support our hypothesis that detection of cfDNA identifies persistently stable levels of MRD as early as day 28 after transplantation and furthermore supports our hypothesis that detection of MRD early after transplantation using this technique identifies patients for whom modifications of the transplant treatment plan are required to reduce the risk of relapse. In contrast, we propose that patients who are MRD-negative in pre-transplant evaluation using cfDNA testing or those patients who clear MRD through the transplant process may be managed without post-transplant consolidation therapy, avoiding the recognized toxicity of such treatment.

We were able to closely correlate the VAF values for the BM and plasma samples, as reported by other laboratories [18,29]. These results support the use of serial cfDNA analyses of plasma samples for disease monitoring without resorting to invasive bone marrow sampling. Such testing as that conducted in this study would appear to be quite valuable in the design and interpretation of transplant trials, including studies involving targeted or non-targeted post-transplant consolidation approaches in which relapse is a designated primary or secondary endpoint. Prospective studies are required to determine if a choice of conditioning or GvHD regimens as well as a donor or cell source may be more effective in clearing adverse-risk somatic mutations through the transplant process. Prospective studies using cfDNA analysis will be required to test our hypothesis that modifications to the post-transplant treatment regimen, such as with consolidation therapy, will reduce the risk of relapse for patients with persistent adverse-risk somatic mutations.

Although our data beyond day 84 are limited, we did observe at least transient clearing of somatic mutations for five subjects and the specific clearing of adverse-risk somatic mutations detected at day 84 for the two patients still in remission, signifying that relapse may not be inevitable for these patients. The clinical relevance of a sustained persistence of low-level epigenetic gene mutations such as DNMT3A and TET2 or other low-risk mutations is uncertain and will require larger studies that can focus on the clinical importance of any individual gene mutation or classes of genes not considered adverse-risk somatic mutations. Longer-term testing of this study population may show that these low-risk mutations may be detected months after transplantation and are evidence of persistence of disease (or CHIP), albeit of uncertain clinical relevance. We suspect that these subjects could suffer relapses in the more distant future if the immunologic graft-versus-leukemia effect of transplantation weakens. The ease of this noninvasive testing, however, facilitates monitoring of such patients with serial testing to address this risk.

The addition of chimerism testing using the same cfDNA technique appeared to provide a warning of an impending relapse, as shown by the decrease in donor chimerism before documentation of a relapse. We note that the BM samples obtained early after transplantation were insensitive to the pending relapse, in contrast to the data developed via plasma cfDNA analysis. We did not find a correlation between the chimerism results determined by NGS of cfDNA and STR analysis of specific cell subsets. Chimerism analysis using cfDNA from plasma samples is expected to reflect the entire body’s DNA, but it has been shown to be significantly more enriched by hematologic cells that are immersed in blood [18]. Accordingly, the results of cfDNA testing may differ from the CD3^+^ or CD15^+^ cell chimerism measurements. The pattern of cfDNA chimerism appears to reflect relapsing disease more accurately than that of CD3 chimerism. All patients who relapsed early after transplantation showed a constant downward trend in cfDNA chimerism, a trend that could be seen even on day 56. Although low or falling donor T-cell chimerism has been associated with a higher probability of relapse [30,31], chimerism testing is not a measure of MRD and does not appear to be useful in identifying individual patients who may benefit from consolidation therapy.

Although not analyzed in this study, mutations can be detected using RNA sequencing. RNA mutation data may add another level of sensitivity for some mutations or chromosomal translocations. We detected multiple variants in the samples obtained before or after transplantation using RNA sequencing (Appendix A). Some of the variants detected may be related to disease, but the expression of multiple unique variants was detected, especially after transplantation, that likely reflect events other than relapse of malignancy. We previously reported that RNA transcriptome analysis of post-transplant bone marrow samples could identify upregulation of genes associated with the occurrence of acute GvHD and with overall survival [32], and such analysis may be helpful in monitoring post-transplant events.

This study is limited by the small number of cases; by selecting subjects with known mutations identified at the time of diagnosis or in subsequent pre-transplant testing; and by the heterogeneity of donor types, conditioning regimens, GvHD prophylaxis regimens, and variable use of post-transplant consolidation, any of which may have affected the clearance of MRD. We used a single commercially available source for the pre- and post-transplant testing for this study, and other NGS platforms may have different sensitivities and specificities or use more limited gene profiles. The clinical importance of most of the genes included in the agnostic panel used in this study is currently not defined and may require larger study populations and long-term follow-ups to determine, including distinguishing between MRD and CHIP.

## 5. Conclusions

Our study explored the use of cfDNA to demonstrate the persistence of disease after transplantation as a possible technique for identifying patients who may benefit from post-transplant consolidation therapy and did not focus on any particular mutation. Based on the non-invasiveness of NGS testing of plasma samples, we propose that serial testing for leukemia-associated cfDNA, especially if combined with chimerism testing using NGS techniques, is an appropriate test for monitoring patients after transplantation for clinically relevant MRD predictive of post-transplant relapses. Such testing allows discussion with the transplant recipient about modification of the individual patient’s treatment plan, with possible improvement in treatment success.

## Figures and Tables

**Figure 1 cancers-17-00625-f001:**
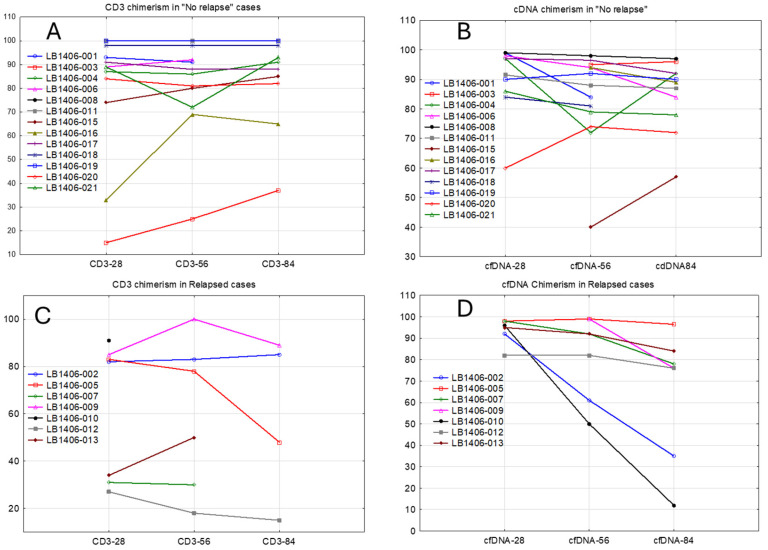
The chimerism values using cfDNA analysis (**B**,**D**) and for CD3+ cells (**A**,**C**) for individual subjects in post-transplant relapse (**C**,**D**) or remission (**A**,**B**).

**Figure 2 cancers-17-00625-f002:**
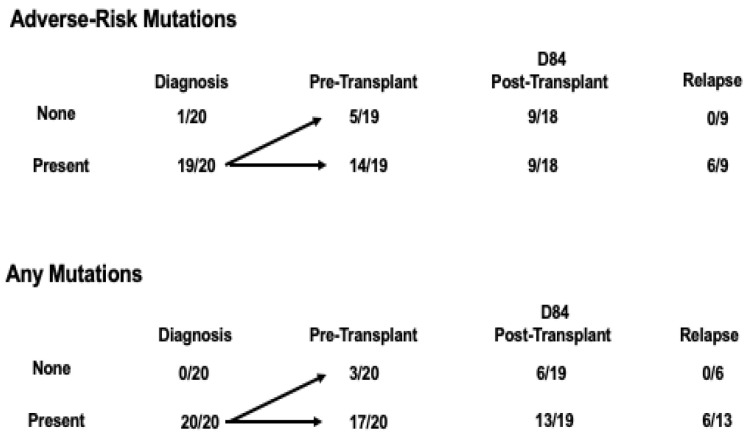
The number of subjects with relapses within 12 months after transplantation, as defined by the presence of adverse-risk or any mutations at various time points before and after transplantation (Table 3). The group listing any mutations includes subjects with adverse-risk or non-adverse-risk mutations.

**Table 1 cancers-17-00625-t001:** Transplant Subject and donor demographics.

Subject Age, Median (Range)	59 (27–77)	Donor Age, Median (Range)	25 (16–56)
Subject Sex (N)		Donor Sex (N)	
Male	13	Male	11
Female	7	Female	9
Transplant Diagnosis (N)		Donor (N)	
Primary AML	6	HLA-Matched Sibling	2
Secondary AML	6	Haploidentical	3
MDS	7	URD Matched	12
CML, Blast Crisis	1	URD Mismatched	3
ABO (N)		CMV (Recipient or Donor, N)	
Match	11	+/+	3
Minor Mismatch	3	+/−	4
Major or Bidirectional Mismatch	6	−/+	4
		−/−	9
HSC Source (N)		Conditioning Regimen (N)	
PBSC	20	MA	6
BM	0	RIC	10
		Non-MA	4
GvHD Regimen (N)			
Tac + MTX	3	Post-Transplant Consolidation (N)	
Tac, MTX + abatacept	11	Yes	9
PTCy	2	No	11
PTCy or abatacept	4		

Subject and donor characteristics are shown. AML = acute myelogenous leukemia; MDS = myelodysplastic syndrome; CML = chronic myelogenous leukemia; GvHD = graft-versus-host disease; CMV = cytomegalovirus status; HLA = histocompatibility locus antigen; URD = unrelated donor; MA = myeloablative; RIC = reduced intensity; Tac = tacrolimus; MTX = methotrexate; PTCy = post-transplant cyclophosphamide.

**Table 2 cancers-17-00625-t002:** Subject and donor demographics and mutations detected through day 84.

Sub No.	DX	Donor/HLA Match	CondReg	GvHD Reg	Post-Transplant Consolidation		Relapse		Current Status	Number of Mutations/Geometric Mean of VAF					
					Regimen	Day Started	Y/N	Day		Pre-Transplant BM	Pre cfDNA	Day 28	Day 56	Day 84	Post-Transplant BM
001	AML	URD 10/10	MA + ATG	MTX	NA		N		Expired, RRT, day 59	3/4.58	2/20.92	5/0.1	3/0.51	ND	ND
002	t-MDS	URD 10/10	MA + ATG	MTX + Abat	NA		Y	170	Alive, >day 365	2/34.42	2/42.72	3/2.4	3/4.02	3/2.77	2/1.13
003	MDS	URD 8/10	NMA	PTCy + Abat	NA		N		Alive, >day 365	2/11.04	0/NE	0/NE	1/2.31	0/NE	0/NE
004	AML	URD 10/10	MA + ATG	MTX + Abat	Sorafenib +HMA	51	N		Expired, infection, day 357	1/44.48	1/44.87	1/0.5	1/0.26	0/NE	1/0.28
005	t-MDS	URD 10/10	RIC + ATG	MTX + Abat	HMA +DLI	121	N		Alive, >day 365	5/3.54	16/0.97	2/0.54	2/0.30	2/0.34	5/0.37
006	2nd-AML (CMML)	URD 10/10	RIC + ATG	MTX + Abat	NA		N		Alive, >day 365	4/37.62	4/35.91	0/NE	0/NE	2/0.11	1/2.55
007	2nd-AML (MDS)	URD 10/10	NMA	PTCy + Abat	NA		Y	62	Expired, relapse, day 250	8/22.97	9/18.72	1/0.66	6/1.07	7/1.52	10/4.29
008	AML	Haplo	RIC	PTCy	Crenolinib +sorafenib	91	N		Alive, >day 365	0/NE	0/NE	0/NE	0/NE	0/NE	0/NE
009	AML	RD 10/10	RIC + ATG	MTX	HMA +MT401 DLI Group 2	76	Y	168	Expired, relapse, day 285	2/0.66	6/0.84	0/NE	0/NE	7/0.9	4/0.24
010	2nd-AML	Haplo	NMA	PTCy + Abat	NA		Y	52	Expired, relapse, day 125	3/8.14	3/23.98	3/0.64	2/19.28	2/33.48	4/5.92
011	2nd-AML (MDS)	URD 9/10	RIC	PTCy + Abat	HMA	88	N		Alive, >day 365	3/19.83	3/9.66	1/1.59	1/3.40	1/6.32	0/NE
012	MDS	URD 10/10	MA + ATG	MTX + Abat	NA		Y	167	Alive, relapse, day 347	4/7.67	11/5.68	5/0.62	3/0.04	7/0.36	4/0.46
013	MDS	URD 10/10	RIC + ATG	MTX + Abat	NA		Y	139	Alive, >day 365	1/48.78	5/2.50	1/0.26	2/1.63	2/1.60	2/4.23
015	AML	RD 10/10	MA	MTX	Midostaurin + gilteritinib	35	N		Alive, >day 365	0/NE	0/NE	1/36.07	1/30.3	1/31.2	2/0.94
016	MDS	URD 10/10	RIC + ATG	MTX + Abat	HMA	89	N		Expired, sepsis, day 268	1/33.18	5/1.46	1/0.98	1/0.03	1/0.06	0/NE
017	CML	URD 10/10	RIC	MTX + Abat	Imatinib	52	N		Alive, >day 365	1/6.66	3/3.78	0/NE	1/0.21	0/NE	0/NE
018	2nd-AML (BrCa)	URD 10/10	RIC	MTX + Abat	NA		N		Alive, >day 365	6/1.42	8/0.63	2/0.57	0/NE	0/NE	ND
019	2nd AML (MF)	Haplo	NMA	PTCy	NA		N		Alive, >day 365	9/5.79	6/9.43	3/0.18	5/0.54	2/0.23	1/0.06
020	AML	URD 10/10	MA + ATG	MTX + Abat	NA		N		Alive, >day 365	5/4.11	7/1.71	1/0.97	1/0.37	0/NE	1/0.37
021	MDS	URD 10/10	RIC + ATG	MTX + Abat	Sorafenib	77	N		Alive, >day 365	2/0.69	4/1.20	0/NE	1/0.11	1/0.11	0/NE

Individual subject and transplant characteristics, post-transplant relapse and survival, and number of mutations and VAF GM at defined time points after transplantation. Host germline mutations (subjects 001, 006, 019, 020, and 021) and donor germline mutations (subjects 002, 004, 017, and 021) were excluded from mutation counts and calculations of VAF GM. Mutations identified via analysis of cfRNA were not included in this analysis. Sub No = subject number; GM = geometric mean; URD = unrelated donor; Haplo = haploidentical related donor without the number of matched alleles shown; HLA = histocompatibility locus antigen; HMA = hypomethylating agent; NA = not applicable; ND = not done; NE = VAF frequency not evaluable; CondReg = conditioning regimens are described by intensity (myeloablative (MA); non-myeloablative (NMA); and reduced intensity (RIC)) with or without rabbit antithymocyte globulin (ATG); GvHD Reg = GvHD prophylaxis was either methotrexate (MTX)-or cyclophosphamide (PTCy)-based, with the addition of abatacept (abat) per physician discretion. Complete details regarding conditioning and GvHD prophylaxis regimens are provided in the Appendix A.

**Table 3 cancers-17-00625-t003:** Comparison of pre-and post-transplant mutations detected in bone marrow and plasma samples.

Sub No	Relapse (Y/N)	Pre-Transplant BM	VAF	Pre-Transplant cfDNA	VAF	Day 84 BM	VAF	Day 84 cfDNA	VAF
001	N	DNMT3a **SRSF2** IDH1	15.85 4.66 1.3	DNMT3A **SRSF2**	28.57 15.38	ND		ND	
002	Y	**TP53** HNF1A	48.34 24.54	**TP53** HNF1A	63.57 28.71	**TP53** HNF1A	2.5 0.51	**TP53** HNF1A	3.19 0.84
003	N	**BCOR** KMT2D	13.85 8.8	Neg		Neg		Neg	
004	N	DNMT3A	44.48	DNMT3A	44.87	DNMT3A	0.28	Neg	
005	N	**TP53** PPM1D CHEK2 NOTCH3 PPM1D	5.2 2.18 7.53 3.95 1.66	**TP53** PPM1D CHEK2 NOTCH3 PPM1D BRAF PPM1D CARD11 KMT2A PMS1 CDK12 PBRM1 **TP53** KEAP1 CARD11 KMT2C	5.53 5.13 4.76 2.79 0.67 5.97 3.23 0.76 0.7 0.58 0.56 0.52 0.35 0.21 0.18 0.11	KMT2A GATA3 MAP3Ki4 TET2 KRCC2	0.19 0.95 0.47 0.37 0.21	**TP53** PPM1D	0.37 0.32
006	N	TET2 **EZH2** **ASXL1** TET2	52.71 47.1 33.71 23.92	TET2 **EZH2** **ASXL1** TET2	49.13 36.36 34 27.4	**ASXL1**	2.55	TET2 TET2	0.25 0.05
007	Y	**SRSF2** **ASXL1** NRAS MTOR MTOR MTOR KDM6A ARAF	55.44 50.62 41.9 31.37 30.0 27.27 6.15 4.17	**SRSF2** **ASXL1** NRAS MTOR MTOR MTOR FH GNAQ IRF4	45.55 48.27 48.59 40.26 38.3 36.09 2.71 4.17 4.11	**SRSF2** **ASXL1** NRAS MTOR MTOR MTOR GNAS GNAS GNAS BCL6	9.4 10.24 7.63 0/73 0.48 0.8 10.41 8.46 8.67 13.43	**SRSF2** **ASXL1** NRAS MTOR MTOR FH DNMT3A	5.62 8.29 4.46 0.55 0.31 1.85 0.27
008	N	Neg		Neg		Neg		Neg	
009	Y	TET2 NRAS	0.28 1.54	TET2 TET2 **WT1** NFKBIA NRAS **FLT3-ITD**	5.78 1.38 0.83 0.6 0.39 0.23	TET2 **WT1** NFKBIA **FLT3-ITD**	0.15 0.69 0.25 0.14	TET2 **WT1** NFKBIA NRAS **FLT3-ITD** **WT1** **WT1**	2.73 3.02 3.77 0.13 2.4 0.12 0.41
010	Y	**SRSF2** MPL IDH2	18.93 4.31 6.61	**SRSF2** MPL IDH2	37.89 23.93 15.2	**SRSF2** MPL IDH2 KMT2C	36.08 0.54 16.24 3.87	**SRSF2** IDH2	31.04 36.11
011	N	**TP53** TET2 PDGFRB	31.54 16.72 14.79	**TP53** TET2 PDGFRB	5.26 8.62 19.86	Neg		PDGFRB	6.32
012	Y	AXIN1 **SF3B1** **ASXL1** **ASXL1** AMER1	16.83 17.7 6.4 4 8.05	AXIN1 **SF3B1** **ASXL1** **ASXL1** AMER1 H3F3A EGFR **RUNX1** KMT2C **ASXL1**	34.8 33.02 3.03 3.01 2.14 17.7 3.46 2.61 1.79 1.4	AXIN1 **SF3B1** **RUNX1** KMT2B	0.25 0.49 0.28 1.68	AXIN1 **SF3B1** AMER1 EGFR **RUNX1** KMT2C KMT2C	0.5 0.35 0.05 0.36 0.28 0.48 1.86
013	Y	Neg		**TP53** TET2 NOTCH1 TET2	43.38 0.78 0.47 0.19	NF2	0.37	**TP53**	0/37
015	N	Neg		Neg		KMT2B DNMT3A	1.25 0.7	Neg	
016	N	**SF3B1**	33.18	**SF3B1** TNFRSF14 KMT2D DNMT3A MAP3K1	43.33 1.48 0.83 0.33 0.26	Neg		**SF3B1**	0.06
017	N	**ASXL1**	6.66	**ASXL1** CEBPA DNMT3A	12.81 5.2 0.81	Neg		Neg	
018	N	ALK **SRSF2** TET2 **SRSF2** DDX41	0.97 0.8 0.51 0.29 1.82	ALK **SRSF2** TET2 **SF3B1** **SF3B1** ALK FGFR4	1.46 0.17 0.2 0.21 0.34 0.44 0.35	ND		Neg	
019	N	CALR **U2AF1** **ASXL1** GNAS KRAS **RUNX1** GALNT12 TP31 GRIN2A	39.23 38.39 34.88 20.25 20.11 2.04 0.87 0.6 0.32	CALR **U2AF1** **ASXL1** GNAS KRAS **RUNX1**	17.76 15.04 19.57 12.98 12.92 0.8	**ASXL1**	0.06	CALR **U2AF1**	0.29 0.18
020	N	DMNT3A. NF1 **ASXL1** TET2 SMC3	32.74 4.4 2.52 3.72 0.87	DMNT3A. NF1 **ASXL1** TET2 SMC3 PDGFRB KMT2C	33.87 4.27 3.64 2.2 0.4 0.4 0.23	DNMT3A	0.37	DNMT3A	0.36
021	N	DNMT3A DNMT3A	1.02 0.47	DNMT3A DNMT3A CBL NFI I	1.38 1.01 1.03 1.46	Neg		CBL	0.11

The gene mutations and VAF values identified in BM or plasma testing before transplant conditioning and at day 84 after transplantation. See Appendix A for complete gene descriptions. All subjects are listed, and relapses within 12 months of transplantation are indicated. Mutations shown in bold type are considered high-risk driver mutations [14,15,16]. Sub No = subject number; ND = not done; Neg = no mutations were identified; VAF = variant allele frequency.

## Data Availability

The human sequence data generated in this study are included within the article and its online Appendix A data files. Further inquiries can be directed toward the corresponding author.

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
