# Peer review of "cfDNA Chimerism and Somatic Mutation Testing in Early Prediction of Relapse After Allogeneic Stem Cell Transplantation for Myeloid Malignancies"

_cancers, 2025, doi:10.3390/cancers17040625_

Round 1
Reviewer 1 Report
Comments and Suggestions for Authors
The authors have proposed a method for predicting recurrence and managing prognosis based on the analysis of cell-free DNA, and have demonstrated the validity and potential of this method through analysis of 21 subjects. This is a useful study, and it is significant that it has been examined.
However, there are some points that need to be revised as an article, so the points for revision are listed below.
1). In Table 3, the presence or absence of recurrence, genetic mutations before and after transplantation, and VAF are listed for all subjects. Although it is useful as a result, it should be presented in a way that clearly supports the main point and argument of the paper.
Specifically, it would be better to include a table or graph that has been re-sorted into two groups according to the presence or absence of recurrence in light of the main point of the paper. In addition, the presence or absence of gene mutations and changes in trends before and after transplantation should be summarized in the form of a graph or other such format.
Perhaps Fig. 1 is a graph that includes some of the above content, but because the resolution of Fig. 1 is extremely low, it was not possible to read the individual characters. As a result, it was not possible to determine whether the appropriate data was graphed in Fig. 1.
2). In the Discussion, the authors make several important suggestions and interpretations. For example, the following three statements
-i). “These low levels of measurable markers indicative of persistent MRD do not necessarily predict for relapse of disease although a lack of clearance of mutations of high-risk driver genes appears associated with a much higher probability of relapse early after transplantation, possibly justifying modifications in the post-transplant treatment regimen for these patients.”
-ii). “The detection of adverse-risk somatic mutations in plasma samples obtained after transplantation identified a specific population with a high risk of relapse”
-iii). “The results of this pilot study support our hypothesis that detection of cfDNA identifies persistently stable levels of MRD as early as day 28 after transplantation”
However, it is difficult to understand which data the arguments are based on. The data supporting these results and arguments should be presented in an easy-to-understand format. For example, by reconstructing and presenting them as graphs and tables corresponding to each argument, the readers will probably understand that the authors' arguments are valid.
In summary, overall, this article is in the form of a list of case reports. Therefore, there is a gap between the authors' arguments and the data. The reader is expected to reconstruct the case data and interpret it for themselves. This is a major cause of the difficulty in reading the article, and it makes it difficult to feel that the validity of the arguments expressed in the abstract and discussion are accurately demonstrated by the data.
Therefore, it is necessary to present the data in a form that allows the reader to analyze it in a logical way that supports the argument, or in a form that allows the reader to easily understand the main point of the data even on first glance, for example in the form of a table.
I believe that the authors' argument is probably correct, but unless some effort is made to make the validity of the argument easily understood by the reader, it will be lacking in the appropriate way to be published as an article.
Author Response
Comment 1). In Table 3, the presence or absence of recurrence, genetic mutations before and after transplantation, and VAF are listed for all subjects. Although it is useful as a result, it should be presented in a way that clearly supports the main point and argument of the paper.
Specifically, it would be better to include a table or graph that has been re-sorted into two groups according to the presence or absence of recurrence in light of the main point of the paper. In addition, the presence or absence of gene mutations and changes in trends before and after transplantation should be summarized in the form of a graph or other such format.
Perhaps Fig. 1 is a graph that includes some of the above content, but because the resolution of Fig. 1 is extremely low, it was not possible to read the individual characters. As a result, it was not possible to determine whether the appropriate data was graphed in Fig. 1.
Response 1) We agree that the data are difficult to follow but chose to include the raw uncensored data for the use by our readers. Accordingly, we added Figure 2, which summarizes the persistence of mutations at day +84 after transplantation for all subjects and for those subjects with adverse-risk mutations and the occurrence of relapse in these groups. We believe this enhances the value of this manuscript.
Comment 2). In the Discussion, the authors make several important suggestions and interpretations. For example, the following three statements
-i). “These low levels of measurable markers indicative of persistent MRD do not necessarily predict for relapse of disease although a lack of clearance of mutations of high-risk driver genes appears associated with a much higher probability of relapse early after transplantation, possibly justifying modifications in the post-transplant treatment regimen for these patients.”
-ii). “The detection of adverse-risk somatic mutations in plasma samples obtained after transplantation identified a specific population with a high risk of relapse”
-iii). “The results of this pilot study support our hypothesis that detection of cfDNA identifies persistently stable levels of MRD as early as day 28 after transplantation”
However, it is difficult to understand which data the arguments are based on. The data supporting these results and arguments should be presented in an easy-to-understand format. For example, by reconstructing and presenting them as graphs and tables corresponding to each argument, the readers will probably understand that the authors' arguments are valid.
In summary, overall, this article is in the form of a list of case reports. Therefore, there is a gap between the authors' arguments and the data. The reader is expected to reconstruct the case data and interpret it for themselves. This is a major cause of the difficulty in reading the article, and it makes it difficult to feel that the validity of the arguments expressed in the abstract and discussion are accurately demonstrated by the data.
Therefore, it is necessary to present the data in a form that allows the reader to analyze it in a logical way that supports the argument, or in a form that allows the reader to easily understand the main point of the data even on first glance, for example in the form of a table.
I believe that the authors' argument is probably correct, but unless some effort is made to make the validity of the argument easily understood by the reader, it will be lacking in the appropriate way to be published as an article.
Response 2) Again, we agree that the manuscript as initially submitted required the reader to pull out the data from the Tables. This is not a series of case reports but rather a prospective study designed to determine if mutations detected after transplantation are clinically relevant. We chose to provide all mutations detected for those readers who would like uncensored data. These data may be helpful to various scientists in their design of future studies to build upon our data. In the Discussion, we refer the reader to the new Figure 2, which supports the statements about which this reviewer is concerned. This Figure 2 provides a summary of the data found in Table 3 and greatly simplifies the reading of the manuscript. We thank the reviewer for this critical comment.
Reviewer 2 Report
Comments and Suggestions for Authors<Overall comments>
The discovery of cfDNA is a long history. Because of lacking good methods with high sensitivity, the relationship of those cfDNA with diseases, especially cancers and infectious diseases, has long been not well established.Within recent two decades, specific fluorescent pigments combined with PCR technique have well developed, particularly those targeting defined mutation in cancer cells. Thus, cfDNA assay now has been widely used in clinical testing for cancer,immune diseases, and prenatal examination.
In this study, the author organized 20 cases of myeloid malignancy as experimental samples to analyze cfDNA before and after allogeneic HSCT. Their data confirm the findings that failure of clearance of MRD after transplantation is associated with disease relapse.
The paper is well written while the method and data convincing. However, the small quantity of clinical cases hamper the scientific level of this study, not mentioning they only observed AML and MDS (only one case of CML) but not other leukemia or lymphoma (in which HSCT are also well administered). As the author showed in Table 3, there were so many mutants involved, only 20 cases could not well answer by which mutation the relapsed cases are related. Besides, since various therapies in leukemia using small molecules or antibodies targeting specific mutants have already widely used clinically, could this cfDNA scanning help in pinpointing a modified drug selection and provide new strategy of therapy is also ambiguous.
<Suggestion>
The author should add current information in introduction and discussion, addressing their cfDNA scanning has some merits and, if sample are expanded, might help in pinpointing mutants with high-rate relapse and developing novel strategy of mutant-targeting therapies.
Author Response
Comments 1) The paper is well written while the method and data convincing. However, the small quantity of clinical cases hamper the scientific level of this study, not mentioning they only observed AML and MDS (only one case of CML) but not other leukemia or lymphoma (in which HSCT are also well administered). As the author showed in Table 3, there were so many mutants involved, only 20 cases could not well answer by which mutation the relapsed cases are related. Besides, since various therapies in leukemia using small molecules or antibodies targeting specific mutants have already widely used clinically, could this cfDNA scanning help in pinpointing a modified drug selection and provide new strategy of therapy is also ambiguous.
The author should add current information in introduction and discussion, addressing their cfDNA scanning has some merits and, if sample are expanded, might help in pinpointing mutants with high-rate relapse and developing novel strategy of mutant-targeting therapies.
Response 1) We agree with these comments regarding the restricted sample. We did state that there was no attempt to compare conditioning or GvHD regimens because such comparisons would require sample sizes of several hundreds, not within the ability of a single site to prospectively study. The small study sample was acknowledged in the Discussion and the necessity of larger sample sizes necessary to dissect out the effects of conditioning regimens, donor selection, post-transplant consolidation, etc.
What our study does show is that 1) cfDNA is detected early after transplantation and 2) the presence of adverse-risk mutations after transplantation is associated with a very high risk of relapse. We proposed in our Introduction and Discussion that such detection very early after transplantation may help select patients at high risk of relapse who may benefit from consolidation to prevent relapse. (We also acknowledged the need for prospective studies to prove any benefit from consolidation therapy.)
In terms of references, in the Introduction we have citations describing the association of somatic mutations with relapse after transplantation (citations 1-4); the possible role of post-transplant consolidation therapy to prevent relapse (citations 5-7); description of cfDNA as a measurement of disease activity (citations 8-12); and the use of cfDNA as a measure of donor cell chimerism (citation 13).
In the Discussion, we have citations discussing the predictive value of minimal residual disease detection (MRD) for post-transplant relapse, including both any mutations and also specific high-risk mutations (citations 17-29). We also have citations discussing various associations of donor cell chimerism with transplant outcomes (citations 31-32).
We believe these citations provide adequate background for our study and support study design and analysis.
Reviewer 3 Report
Comments and Suggestions for Authors
The authors investigate the role of cfDNA in chimerism and MRD assessment in a single institution series of patients undergoing allogeneic HSCT because of myeloid neoplasms. They conclude that cfDNA monitoring could be a useful tool to early identify patients at risk of relapse, so allowing tailored post-transplant treatment to be delivered.
This pilot study has a prospective design. On the other hand, the series is limited and heterogeneous. Nevertheless, the authors properly present their research and thoroughly discuss strengths and weaknesses. Therefore, the paper is highly appreciable and the conclusions can be easily shared.
I have only two minimal, possibly pedantic, remarks, by no means affecting the relevance of the manuscript.
1) I am old enough to remember what “consolidation therapy” originally meant. The authors collect under this heading an array of treatments, either disease specific or not, either pharmacologic or immunotherapeutic, either licensed or off label. I wonder whether “consolidation therapy” is the optimal term and whether the matter could be better presented.
2) In the supplement: regiments instead of regimens; moreover, there is some possible spacing error: e. g. “ MA consists”.
Author Response
Comments 1) I am old enough to remember what “consolidation therapy” originally meant. The authors collect under this heading an array of treatments, either disease specific or not, either pharmacologic or immunotherapeutic, either licensed or off label. I wonder whether “consolidation therapy” is the optimal term and whether the matter could be better presented.
Response 1) We believe "consolidation" is an appropriate term. This is in contrast to maintenance, which implies long-term therapy that is not usually given after allogeneic HSC transplantation, or "intensification," which is not used after allogeneic transplantation. We did state that both targeted and non-targeted therapies were given.
Comment 2) In the supplement: regiments instead of regimens; moreover, there is some possible spacing error: e. g. “ MA consists”.
Response 2) We found the typographical errors and these were corrected in the revised supplement that is now uploaded into the system.
Round 2
Reviewer 1 Report
Comments and Suggestions for Authors
I confirmed that appropriate measures had been taken, such as the addition of a figure summarising the overview.